# MED-EL hearing solution registry: An examination of the strengths and limitations of a cochlear implant registry

Uwe Baumann[1]*, Tobias Weissgerber[1], Andreas Radeloff[2], Karin A. Koinig[3], Magdalena Breu[3], Jasmine Rinnofner[3], Vera Lohnherr[4], Stefan Dazert[5], Christiane Völter[5], Ilona Anderson[3], Joachim Müller[6], Daniel Polterauer[6]

1 ENT/Audiological Acoustics, University Hospital, Goethe University Frankfurt, Frankfurt am Main, Germany, 2 Division of Otorhinolaryngology and Research Center Neurosensory Science, Carl von Ossietzky University Oldenburg, Oldenburg, Germany, 3 Clinical Research Department, MED-EL Elektromedizinische Geräte GmbH, Innsbruck, Austria, 4 Hals-, Nasen- und Ohrenklinik (Kopfklinik) Im Neuenheimer Feld, University Hospital Heidelberg, Heidelberg, Germany, 5 Department of Otorhinolaryngology, Head and Neck Surgery, St. Elisabeth Hospital, Ruhr University, Bochum, Germany, 6 Clinic and Polyclinic for Ear, Nose, and Throat Medicine, Ludwig Maximilian University of Munich Hospital, Munich, Germany

* Uwe.Baumann@unimedizin-ffm.de

## Abstract

There is a significant gap in routine clinical data concerning cochlear implant (CI) users, particularly regarding real-world outcomes. To address this, the present study focuses on the MED-EL Hearing Solution (MEHS) multicenter registry, which covers 5 clinics in Germany. Participants were users of a SONNET 2 or SONNET 2 EAS audio processor. Anonymized data that were routinely collected were extracted and analyzed by a third-party organization. Speech test outcomes (monosyllables and numbers), hours of daily use, self-perceived level of auditory benefit, and safety were assessed. Outcomes from 361 SONNET 2 (or SONNET 2 EAS) users, including 57 bilateral users, were extracted and analyzed. Speech test outcomes increased significantly from pre-operatively to 1 year of CI use in monosyllables (median 10.0% to 65.0%) and numbers (median 77.8% to 100.0%) (both p < 0.001). The majority (79%) of people used their device for at least 9 hours a day. The level of self-perceived auditory benefit was "moderate". Safety was assessed as per participant self-report: 17 minor clinical events were reported, none were new or unresolved. The data obtained from the MEHS registry offers valuable real-life evidence from routine clinical evaluations, which can support clinicians in developing more informed guidelines and treatment plans. The SONNET 2 (EAS) is effective and safe to use. Registries have the potential to provide a wealth of data that can be used to improve hearing health care. However, the success of registries depends on consistent participation and adherence to standardized protocols by clinical centers. Variability in clinical

**Data availability statement:** All relevant data are within the paper and its Supporting Information files.

**Funding:** This study was funded by MED-EL Elektromedizinische Geräte G.m.b.H. The funders assisted in study design, data collection and analysis, and preparation of the manuscript. Decision to publish was a joint decision between all authors.

**Competing interests:** The authors would like to disclose that Karin A. Koinig, Magdalena Breu, Jasmine Rinnofner, and Ilona Anderson are employed at MED-EL. All other authors declare no conflicts of interest. This study was designed in collaboration with MED-EL.

practices, demographic factors, and reporting standards across centers poses significant challenges to achieving homogeneous and usable data.

## Introduction

Hearing loss profoundly impacts individuals and society, particularly as it becomes more prevalent with advancing age [1–7]. Cochlear implantation is the standard of care for individuals with permanent unilateral or bilateral sensorineural hearing loss who derive insufficient benefit from hearing aids [8,9]. However, despite technological advancements, few "best-practice" guidelines exist to inform professionals about the use of implantable hearing devices [10]. Instead, various implant programs have developed individual protocols, each with their indications based on clinician expertise, published literature, and jurisdiction-specific services. The available data to support uniform evidence-based practice remains limited and heterogeneous because relatively few studies that aimed at collecting long-term patient-related outcomes have been conducted on large, representative population samples [11]. As healthcare systems and budgets face increasing pressure, there is a growing demand for evidence-based research that demonstrates the safety and effectiveness of interventions in real-world environments across diverse populations.

Several authors have advocated using standard electronic registries to collect homogeneous data sets for CI users [12–17]. Prospective, longitudinal patient outcome registries have been proposed as an effective means of addressing the increasing demand for data on the benefits of medical devices, as emphasized by the United States Food and Drug Administration (FDA) [12,13]. Moreover, these electronic registries can provide valuable data demonstrating improvements in both speech perception and quality of life [18]. However, implementing such a registry for large and diverse user groups presents significant challenges, including cost and complexity [19]. Currently, there is a lack of routine clinical data on the long-term outcomes of individuals with hearing implants, such as CI users. Routine clinical data is crucial for enhancing our understanding of diseases, evaluating the clinical effectiveness of interventions, improving the quality of care, assessing medical device effectiveness, and determining healthcare outcomes and socio-economic impacts. As our ability to connect data sources and build "big" data increases, registry information could play a pivotal role in measuring the outcomes of healthcare investments and guiding policy and decision-making. To this end, it is essential to create a registry for individuals using various types of implantable hearing solutions. To address this need, a large, prospective, multicentre registry of patient outcomes for CI users of all ages has been implemented in Germany.

This study highlights the use of the MED-EL Hearing Solutions (MEHS) registry, with the primary aims of 1) facilitating consistent and reliable cross-cultural data collection across all age groups via an electronic platform, and 2) gathering extensive longitudinal datasets to provide evidence of device effectiveness. The secondary aim of this investigation was to collect routine clinical data on the performance, effectiveness, and safety of the SONNET 2 and SONNET 2 EAS audio processors.

## Methods

### Ethics approval and informed consent

Ethical approval was obtained by the ethics committees of the Ludwig Maximilian University of Munich Hospital (Ethics number: #17-227), Goethe-University Frankfurt (Ethics number: #288/17), and University of Oldenburg (Ethics number: #2018-110), University Hospital Heidelberg (Ethics number: #S-546/2018), and the Ruhr University of Bochum (Ethics number: #17-6152). The MEHS Registry is performed in adherence with the standards set in the latest revision of the Declaration of Helsinki. The MEHS Registry was registered within the clinical trials database (ClinicalTrials.gov: NCT05668338).

All CI users provided written informed consent before participating in the registry.

### Registry cohort and visits

The MEHS Registry prospectively collects non-interventional clinical data according to clinical routine in children and adults with one or more MED-EL hearing devices. Data are collected as fully anonymized data sets, derived from original clinical records on appropriately informed subjects. The data are implemented into the nationwide MEHS Registry through a secure, web-based platform that enables the collection of data from clinicians and CI users at clinically consistent time intervals. For this evaluation, MEHS Registry data was extracted in May 2023 (Munich on 9-May-2023, Heidelberg on 11-May-2023, Frankfurt on 15-May-2023, and Oldenburg and Bochum on 31-May-2023), aiming at analyzing longitudinal changes in speech perception, subjective perception of sound quality, and wearing time in SONNET 2 or SONNET 2 EAS users. Note, SONNET 2 and SONNET 2 EAS were not differentiated in this study; as such, the term "SONNET 2 (EAS)" encompasses both. Due to the observational nature of the MEHS Registry, no specific visit schedule was defined. Observational data were collected and assigned to baseline (0 to <12 months pre-surgery) or follow-ups: 3 months (0 to <3 months), 6 months (3 to <12 months), and 1 year (12 to <24 months) post-implantation. This enabled the comparison of results over time.

The registry's inclusion criteria encompass both children and adults who received a non-implanted or implanted MED-EL device uni- or bilaterally. To be included in this study, users had to have 1) received a SONNET 2 (EAS) audio processor, 2) pre- and post-operative speech perception data (at least one post-operative interval), 3) sound quality assessment and wearing time data at least one post-operative interval, and 4) received a known electrode array (i.e., this information must not be missing for analyzing the Freiburg monosyllabic speech test and the Freiburg number test). The registry's exclusion criteria included anything that would expose the participant to an increased risk or that would prevent the CI user's full compliance with the completion of the MEHS Registry's procedures. For this study, the only specific exclusion criterion was failure to meet all the inclusion criteria.

### The SONNET 2 (EAS)

SONNET 2 introduced improvements in sound processing algorithms, automatic environment detection, and wireless connectivity. Unlike SONNET 1, SONNET 2 features Adaptive Intelligence for better automatic scene classification and dual microphone directionality, which enhances speech understanding in complex acoustic environments [20,21]. It also supports direct audio streaming via AudioLink, improving user experience and device usability. These enhancements warrant a dedicated evaluation, as they may influence real-world hearing outcomes, user satisfaction, and long-term device performance.

### Measurements of performance, effectiveness, and safety in the SONNET 2 (EAS)

**Speech perception in quiet: Freiburg speech tests.** Speech perception in quiet was assessed via the Freiburg speech tests. Testing was conducted unilaterally in the best-aided condition. If participants had useable contralateral hearing, that ear was occluded or masking was applied. Pre-surgery assessments while aided with a CI (re-implantation) were excluded. The tests were performed at 65 dB SPL in a soundproof room in free-field conditions. The speaker was positioned at head-level (0° azimuth) 1 m from the CI users. In the Freiburg monosyllabic speech (FMS) test, the CI

user was presented with a recorded list of 20 monosyllabic words. In the Freiburg Number (FNum) test, the CI user was presented with a recorded list of 10 four-syllable numbers. The CI user was then asked to repeat the word they perceived and was scored by the percent of words (FMS) or correct numbers (FNum) [22,23].

**Wearing time.** The audio processor wearing time was determined using clinical data logging. However, if clinical data logging was not possible, the wearing time reported by the CI user was recorded.

**Subjective sound quality: hearing implant sound quality index.** The subjective sound quality in the CI-users' personal, everyday listening situation with their hearing implant was assessed with the Hearing Implant Sound Quality Index (HISQUI19). The Hearing Implant Sound Quality Index (HISQUI19) consists of 19 items scored on a 7-point Likert scale in which 7 is "always" and 1 is "never". The total score is the sum of the individual item scores. Total scores are assigned a qualitative level of quality of sound: 110–133 is "very good", 90–109 is "good", 60–89 is "moderate", 30–59 is "poor", and <29 is "very poor" [24].

**Safety.** The safety of the SONNET 2 (EAS) was assessed by collecting clinical events. Within the context of this registry, such events were monitored on-site and reported according to the incident reporting procedure for approved products. Device deficiencies occurring within the context of this registry were recorded and monitored on-site. Device deficiencies were classified based on the severity of the issues to the device and their potential impact on patients' safety. Severity classifications were determined by reported patient experiences and by the anticipated clinical implications of the device deficiency, based on device performance characteristics and clinical risk profiles.

## Equipment and software

**ENTstatistics.** ENTstatistics (Innoforce, Ruggell, Liechtenstein) is a database program designed to help ENT specialists manage the clinical data of CI users. ENTstatistics allows users to document treatment data, such as surgical reports, audiograms, and speech test results, as well as the results of other audiological diagnostics, follow-up reports, sketches, and images in a logically structured form. The therapy data are then stored in a database that can be searched and processed based on many different criteria. ENTstatistics can be connected to an existing hospital information system (HIS). An interface integrated into the HIS system allows ENTstatistics to be called to display the treatment data provided for a patient. The methodological schematic of the data extraction using ENTstatistics is shown in Fig 1. The ENTstatistics software was used to collect data at the participating clinics of the present study. The pooled, anonymized data were then analysed within MED-EL headquarters. This software allows for the simple and rapid documentation and processing of data for the diagnosis and treatment of CI users.

**Multi-center study system.** The Multi-Centre Study (MCS) web application is the Web-based portion of the MEHS Registry used for the combined collection and analysis of treatment data. Through this application, the MEHS Registry sponsor (MED-EL) downloaded the merged, anonymous data files of all participating clinics. The anonymized datasets were therefore provided to the authorized third party (MED-EL) in response to a specific query. Finally, all anonymized data files were imported into a local copy of ENTstatistics for global evaluation.

## Data extraction, monitoring, and analysis

**Data extraction.** Following a pre-defined extraction protocol and inclusion/exclusion criteria, MED-EL extracted the data used for all interim analyses. All extracted data were anonymous to data managers and statisticians, working within the clinical research department (MED-EL) and thus pooled for analyses. All extracted data, in the current manuscript, focused on SONNET 2 (EAS) audio-processor users. To implement the MEHS Registry data extraction, the participating hospitals have locally installed ENTstatistics software (MCS edition), which can be directly integrated into clinical practice. The software is managed under the responsibility of the clinical center and embedded in the clinical routine. MED-EL accessed fully anonymized data upon query determination, according to the MEHS Registry protocol through the MCS server. To evaluate therapy data, a local copy of ENTstatistics with evaluation functions was used by the MEHS Registry sponsor (MED-EL).

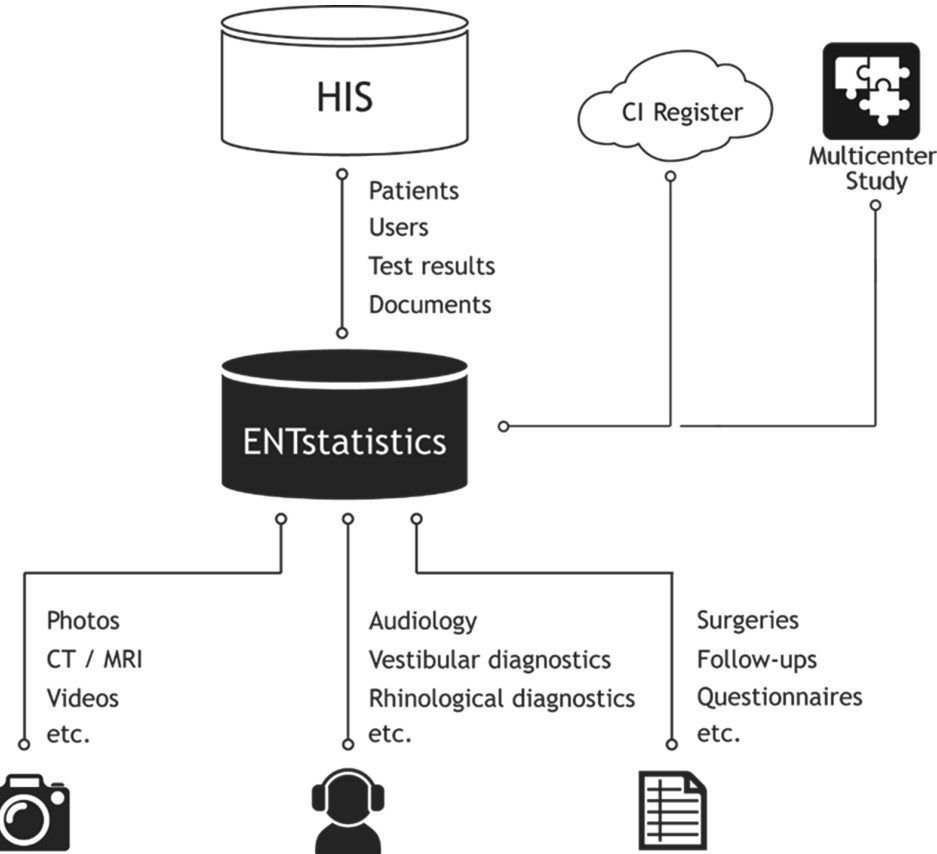

**Fig 1. Methods schematic: Schematic of ENTstatistics® interfacing with the hospital information system (HIS) and other data sources.** From the ENTstatistics, the data is then sent to the MED-EL Hearing Solutions (MEHS) cochlear implant (CI) Register and Multi-Centre Study (MCS) server. The MEHS Register and MCS server comprised the web-based portion of the registry used for the combined collection and analysis of therapy data. This figure was created by Christoph Wille from Innoforce.

**Data monitoring.** The conduct of this registry was monitored according to a monitoring plan. Pre-investigation monitoring was conducted, on-site, after the approval of the MEHS registry by the relevant ethical committee and in advance of the MEHS registry initiation. The pre-investigation monitoring aimed to prepare the investigation site for the conduct of the registry. During this visit, the investigator site file was completed with all necessary documents, and key members of the investigational team were trained. Periodic monitoring is conducted throughout the MEHS data collection. The data management plan guides data collection, therefore ensuring that all data collected are within accurate and reliable standards. The data managers were responsible for guaranteeing the correct extraction and organization of the anonymized data, as per the clinical investigation plan.

## Statistical analysis

Descriptive statistics were used to report subject characteristics and to provide a summary of the examined study outcomes. All quantitative data are presented as mean with standard deviation (SD) and/or median with interquartile range (IQR), minimum, and maximum; qualitative data are presented as absolute and relative frequencies. In box plots, the boxes enclose the interquartile range (middle 50% of data), the horizontal lines represent the median, the Xs represent the mean, and the whiskers separate the outliers, which are shown as circles. Quartile calculation includes medians.

To assess improvements in FMS and FNum, we considered the following: Registry participants do not follow a pre-specified visit schedule and are not evaluated at every predefined period. Therefore, we restricted the comparison to those with a pre-surgical (baseline) and at least one post-surgical assessment. Thus, we used any data reported for the predefined post-surgical periods, regardless of whether the SONNET 2 (EAS) user was assessed during the previous period. Consequently, we do not compare intra-individual changes. Missing data were treated as missing values. The data followed a non-parametric distribution, with FNum and FMS scores reaching a ceiling post-surgery. This was confirmed by the Shapiro-Wilk test and visual inspection of Q-Q plots. Therefore, we used the Mann-Whitney U test (MWU) for unpaired data and reported median and IQR (unless n < 4, for which we reported minimum and maximum). We restricted the statistical comparison of the FMS and FNum scores to pre-surgical (baseline) versus 1 year (12 to <24 months) post-surgical, with at least 10 assessments available at the 1-year follow-up. Changes within the test-retest variability were considered clinically irrelevant, that is, ±10% for FMS and ±20% for FNum. A two-sided p < 0.05 was considered statistically significant. All analyses were performed using SPSS (IBM SPSS Statistics version 29).

## Results

### Cochlear implant user demographics

CI user demographics are presented in Table 1. Fig 2 demonstrates a consort diagram for the study participation. The current study initially extracted data from 369 SONNET 2 (EAS) users (Table 1; Fig 2). Of the initial 369 users, 57 were using a SONNET 2 (EAS) bilaterally (Table 1). The proportion of electrode arrays and implants used in

**Table 1. Cochlear implant user demographics: FMS = Freiburg monosyllable speech test, FNum = Freiburg number test.**

| | All | | FMS | | FNum | | Wearing Time | |
|---|---|---|---|---|---|---|---|---|
| Demographics | n | % | n | % | n | % | n | % |
| **Participants** | **369** | | **91** | | **88** | | **151** | |
| ears | 426 | | 96 | | 94 | | 159 | |
| bilaterally implanted | 57 | | 5 | | 6 | | 8 | |
| **Gender** | | | | | | | | |
| female | 182 | 49.3 | 48 | 52.7 | 48 | 54.5 | 73 | 48.3 |
| male | 187 | 50.7 | 43 | 47.3 | 40 | 45.5 | 78 | 51.7 |
| other | 0 | 0.0 | 0 | 0.0 | 0 | 0.0 | 0 | 0.0 |
| **Ear to be implanted** | | | | | | | | |
| left | 208 | 48.8 | 43 | 44.8 | 46 | 48.9 | 74 | 46.5 |
| right | 218 | 50.9 | 53 | 55.2 | 48 | 51.1 | 84 | 52.8 |
| **Age (years)** | | | | | | | | |
| <5 | 33 | 8.9 | 4 | 4.4 | 4 | 4.5 | 28 | 18.5 |
| 5 to <18 | 22 | 6.0 | 13 | 14.3 | 13 | 14.8 | 21 | 13.9 |
| 18 to <40 | 50 | 13.6 | 29 | 31.9 | 29 | 33.0 | 46 | 30.5 |
| 40 to <60 | 120 | 32.5 | 36 | 39.6 | 33 | 37.5 | 47 | 31.1 |
| 60 to <80 | 124 | 33.6 | 9 | 9.9 | 9 | 10.2 | 9 | 6.0 |
| >=80 | 20 | 5.4 | | | | | | |
| **Age** | yr | | yr | | yr | | yr | |
| median | 55.3 | | 59.8 | | 59.6 | | 55.7 | |
| mean | 50.1 | | 58.2 | | 57.7 | | 48.6 | |
| SD | 23.7 | | 18.2 | | 18.4 | | 25.0 | |
| minimum | 0.6 | | 12.2 | | 12.2 | | 0.6 | |
| maximum | 88.7 | | 88.7 | | 88.7 | | 86.8 | |

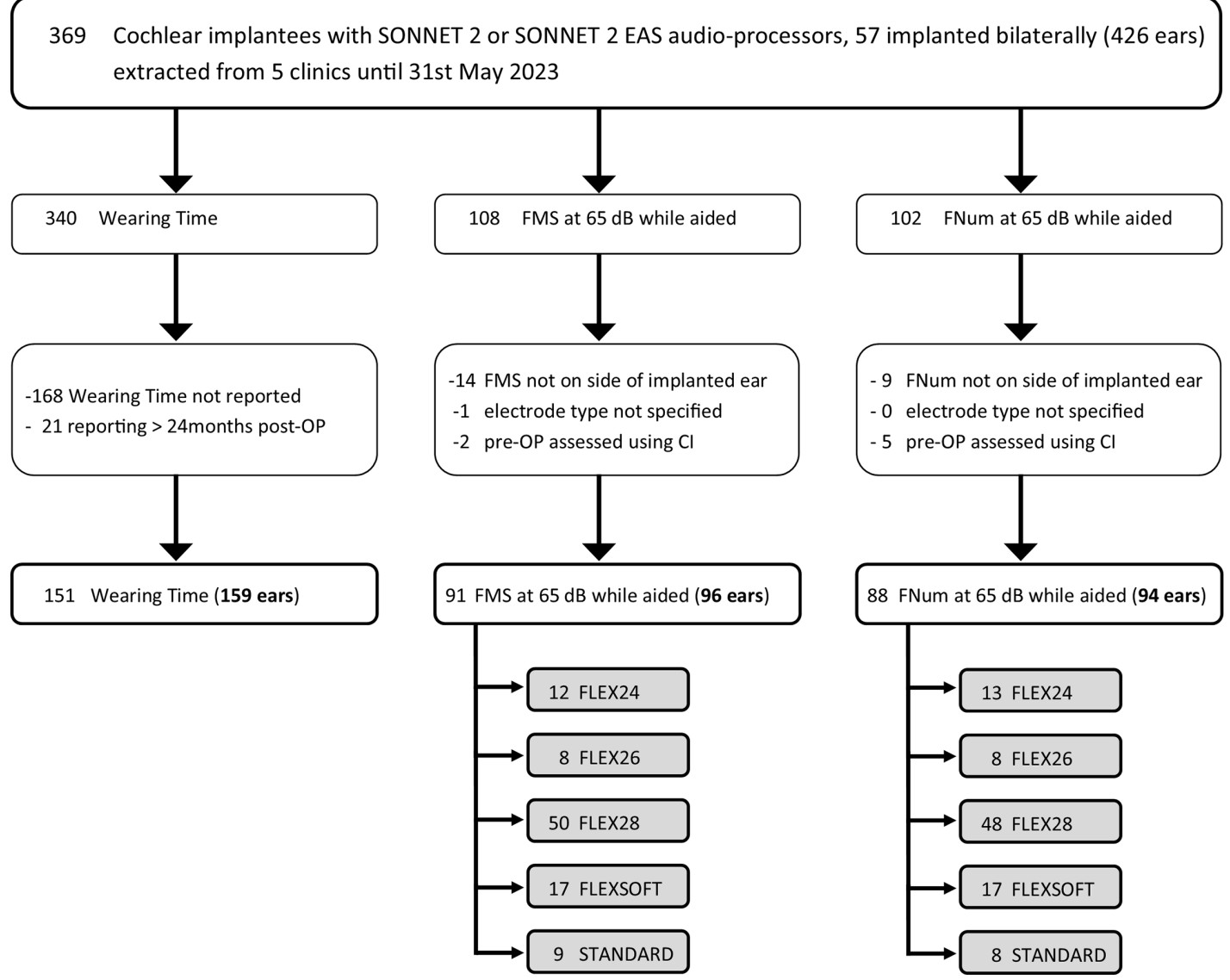

**Fig 2. CONSORT diagram of the MED-EL Hearing Solutions (MEHS) registry.** Number of CI users in the total extraction and the sub-cohorts with Wearing Time, Freiburg monosyllabic speech (FMS), and number (FNum) tests. Includes assessments and exclusion steps. Note that electrode array types (grey shaded boxes) are listed per ear.

SONNET 2 (EAS) users is presented in Table 2. We found that most CI users in the database used the FLEX28 (34%), FLEXSOFT (27%), and STANDARD (29%) electrode arrays, while a small percentage used the FLEX24 (4%) and FLEX26 (6%) (Table 2). Moreover, we saw that most CI users in the database use the SYNCHRONY 2 implant (30%) or the type of their MED-EL implant was not specified (28%) with all the other iterations having less use (SYNCHRONY, 16%; SYNCHRONY PIN, 4%; SYNCHRONY 2 PIN, 4%; CONCERTO, 11%; CONCERTO PIN, 0%; SONATA, 2%; PULSAR, 2%; C40 + , 3%) (Table 2). There were datasets for 91 CI users for the FMS test and datasets for 88 CI users for the FNum (Table 1; Fig 2). 151 CI users are included in the evaluated overall wearing time (n = 151).

**Table 2. Proportion of electrode arrays and implants used in SONNET 2 (EAS) users and in the sub-cohorts. FMS = Freiburg monosyllable speech test, FNum = Freiburg number test.**

| Devices used | All | | FMS | | FNum | | Wearing Time | |
|---|---|---|---|---|---|---|---|---|
| | n | % | n | % | n | % | n | % |
| ears | 426 | | 96 | | 88 | | 159 | |
| **Electrodes** | | | | | | | | |
| FLEX24 | 16 | 3.8 | 12 | 12.5 | 13 | 13.8 | 5 | 3.1 |
| FLEX26 | 11 | 2.6 | 8 | 8.3 | 8 | 8.5 | 7 | 4.4 |
| FLEX28 | 145 | 34.0 | 50 | 52.1 | 48 | 51.1 | 54 | 34.0 |
| FLEXSOFT | 113 | 26.5 | 17 | 17.7 | 17 | 18.1 | 48 | 30.2 |
| STANDARD | 123 | 28.9 | 9 | 9.4 | 8 | 8.5 | 42 | 26.4 |
| MEDIUM | 1 | 0.2 | 0 | 0.0 | 0 | 0.0 | 1 | 0.6 |
| COMPRESSED | 0 | 0.0 | 0 | 0.0 | 0 | 0.0 | 0 | 0.0 |
| FORM19 | 0 | 0.0 | 0 | 0.0 | 0 | 0.0 | 0 | 0.0 |
| unknown | 17 | 4.0 | 0 | 0.0 | 0 | 0.0 | 2 | 1.3 |
| **Implants** | | | | | | | | |
| SYNCHRONY | 70 | 16.4 | 24 | 25.0 | 24 | 25.5 | 6 | 3.8 |
| SYNCHRONY PIN | 15 | 3.5 | 1 | 1.0 | 0 | 0.0 | 6 | 3.8 |
| SYNCHRONY 2 | 128 | 30.0 | 47 | 49.0 | 46 | 48.9 | 97 | 61.0 |
| SYNCHRONY 2 PIN | 18 | 4.2 | 1 | 1.0 | 0 | 0.0 | 12 | 7.5 |
| CONCERTO | 45 | 10.6 | 4 | 4.2 | 5 | 5.3 | 1 | 0.6 |
| CONCERTO PIN | 1 | 0.2 | 0 | 0.0 | 0 | 0.0 | 0 | 0.0 |
| SONATA | 7 | 1.6 | 1 | 1.0 | 2 | 2.1 | 0 | 0.0 |
| PULSAR | 9 | 2.1 | 2 | 2.1 | 2 | 2.1 | 0 | 0.0 |
| C40+ | 13 | 3.1 | 0 | 0.0 | 0 | 0.0 | 0 | 0.0 |
| unknown | 120 | 28.2 | 16 | 16.7 | 15 | 16.0 | 37 | 23.3 |

## Speech testing

**Freiburg monosyllabic speech test.** The outcomes of the FMS test of 91 SONNET 2 (EAS) users are shown in Fig 3. For the data summarized across all electrode arrays (Fig 3A), our analysis showed a significant improvement in the FMS test score from pre-surgical baseline (10.0%, IQR 0.0–29.4) to 6-months post-surgery (65.0%, IQR 50.0–80.0%; MWU $U = 33.50$, $Z = -7.451$, $p < 0.001$) and to 1-year post-surgery (65.0, IQR 51.3–75.0%; MWU $U = 61.00$, $Z = -6.714$, $p < 0.001$) (Table 3).

When assessing each electrode array separately (Fig 3B; Table 3), we generally also observed an improvement in FMS test scores from pre-surgery to after 1-year post-surgery (Fig 3B), but inclusion numbers were too low for inferential statistical assessments except for the FLEX28. The scores for each array were: in FLEX24 from pre-surgery (n = 11, 30.0%, IQR 15.0–35.0%) to 1-year post-surgery (n = 4, 72.5%, IQR 63.8–80.0%); in FLEX26 from pre-surgery (n = 5, 5.0%, IQR 5.0–10.0%) to 1-year post-surgery (n = 4, 83.8%, IQR 76.9–90.0), in FLEX28 from pre-surgery (n = 24, 6.7%, IQR 0.0–20.0%) to 1-year post-surgery (n = 17, 65.0%, IQR 55.0–70.0%; MWU $U = 8.00$, $Z = -5.242$, $p < 0.001$), in FLEXSOFT from pre-surgery (n = 6, 10%; IQR 1.3–15.0%) to 1-year post-surgery (n = 3, min = 20%, max = 70%). In the STANDARD array, pre-surgery assessments were absent, and only two users were assessed 1 year post-surgery. While scores increased from 3 months (n = 5, 10%, 5.0–15.0%) to 1-year post-surgery (n = 2, min = 20%, max = 35%), compared to those assessed at 6-months post-surgery (n = 3, min = 50%, max = 75%), the two SONNET 2 (EAS) users assessed at the 1-year follow up scored lower. However, a higher number of SONNET 2 (EAS) users implanted with STANDARD, and an FMS test would be necessary to properly evaluate this pattern.

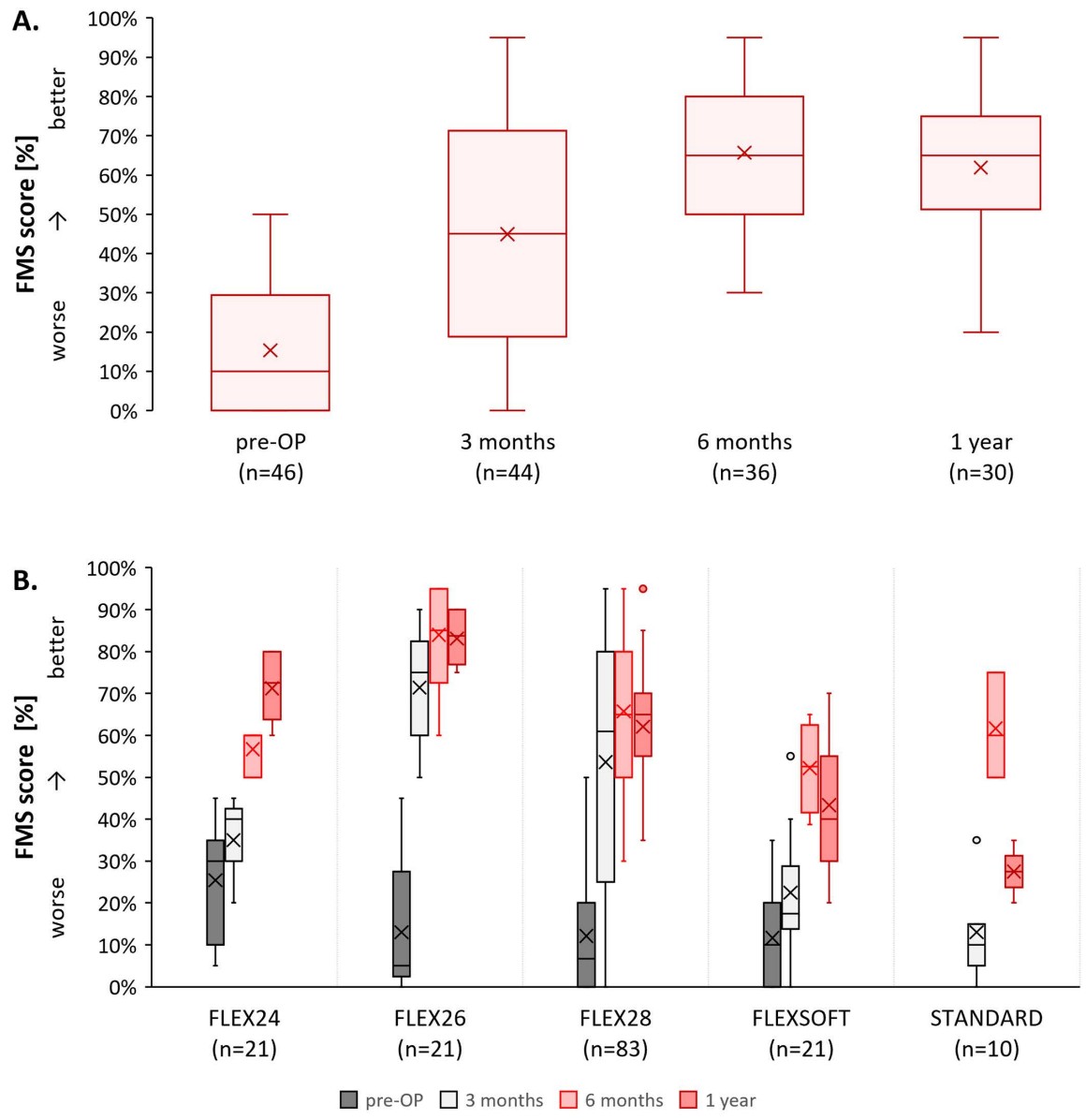

**Fig 3. Results of Freiburg monosyllabic speech scores [%] stratified by A) interval and B) electrode array and interval.** Small circles indicate outliers.

## Freiburg number test

The outcomes of the FNum test of 88 CI users (n = 88) are shown in Fig 4. For the data summarized across all electrode arrays, there was a significant change in the FNum test from pre-surgical baseline (77.8%, IQR 40.0–90.0) to 6-months post-surgery (100.0%, IQR 90.0–100.0; MWU $U = 278.00$, $Z = -4.300$, $p < 0.001$) and 1-year post-surgery (100.0%, IQR 95.0–100.0; MWU $U = 87.50$, $Z = -3.380$, $p < 0.001$) while using the SONNET 2 (EAS) (Fig 4A; Table 3). As expected, and in line with the clinical routine to use this test early due to its simplicity, we found that fewer FNum tests are available for later follow-up. When assessing each electrode array separately,

**Table 3. Freiburg tests: FMS: Assessed in 91 different individuals (96 ears). FNum: Intelligibility (%) of two-digit numbers, compared per period in 94 different individuals. N = number of tests per period per ear; SD = Standard deviation; IQR = Interquartile range.**

| Monosyllables (FMS) | Pre-surgery | 3-months | 6-months | 1-year |
|---|---|---|---|---|
| n | 46 | 44 | 36 | 30 |
| Mean (%) | 15.3 | 44.9 | 65.7 | 61.9 |
| SD (%) | 15.8 | 29.4 | 17.7 | 20.1 |
| Median (%) | 10.0 | 45.0 | 65.0 | 65.0 |
| IQR (%) | 0.0–29.4 | 18.8–71.3 | 50.0–80.0 | 51.3–75.0 |
| Minimum (%) | 0.0 | 0.0 | 30.0 | 20.0 |
| Maximum (%) | 50.0 | 95.0 | 95.0 | 95.0 |
| Numbers (FNum) | Pre-surgery | 3-months | 6-months | 1-year |
| n | 45 | 52 | 29 | 11 |
| Mean (%) | 63.1 | 81.5 | 95.2 | 97.3 |
| SD (%) | 35.8 | 24.2 | 6.9 | 4.7 |
| Median (%) | 77.8 | 90.0 | 100.0 | 100.0 |
| IQR (%) | 40.0–90.0 | 70.0–100.0 | 90.0–100.0 | 95.0–100.0 |
| Minimum (%) | 0.0 | 0.0 | 80.0 | 90.0 |
| Maximum (%) | 100.0 | 100.0 | 100.0 | 100.0 |

we generally also observe an improvement in FNum test scores from pre-surgery to after 1-year post-surgery (Fig 4B), with a distinct ceiling effect due to the simplicity of this test. However, the 1-year assessments were only available for one or two SONNET 2 (EAS) users for FLEX24, FLEX26, and FLEXSOFT, making it difficult to draw definitive conclusions. We found that with FLEX24 the FNum scores started from a very high level pre-surgical (n = 10, 90%, IQR 89%–89%), increased at 3-months (n = 4, 95%, IQR 70–70%) but with very high variability, peaked at 6-months (n = 3, min = 90%, max = 100%), and plateaued at 1 year (n = 2, min = 90%, max = 100%) post-surgery (Fig 4). FNum scores in the FLEX26, FLEX28, and FLEXSOFT consistently increased from pre-surgery to post-surgery: FLEX26: n = 4, 70%, IQR 50.0–50.0% to n = 4, 100.0%, IQR 100.0–100.0% at 6 months; FLEX28: n = 26, 70%, IQR 33%–33.0% to n = 5, 100.0%, IQR 90.0–90.0% at 1 year; and FLEXSOFT: n = 5, 20%, IQR 0.0–0.0% to 90%, IQR 100.0–100.0% at 6 months.

Inclusion numbers were too low for statistical assessments except for FLEX28 at 6 months post-surgery ($U = 92.50$, $Z = -3.323$, $p < 0.001$). With the STANDARD array, the pre-surgical assessment was absent, and only one user was assessed 6 months (n = 1; 90%) and 1 year post-surgery (n = 1; 100%).

## Wearing time

For 159 SONNET 2 (EAS) audio processors in use (151 CI users, 159 ears), wearing time was reported and summarized for up to 1 year of device use. The last reports of the wearing time of each ear were included in the summary. The majority (58%) used their CI for over 12 hours per day; 21% used it 9–12 hours per day, 7% used it 6–9 hours per day, 11% used it 3–6 hours per day, and 3% used it 0–3 hours per day.

## Self-perceived auditory benefit

Our analysis showed only a slight increase in the HISQUI19 scores from the pre-surgical baseline (64 ± 24 score, n = 54) to 1-year post-cochlear implantation (74 ± 21 score, n = 17) (Fig 5). Although both the average and median scores from pre-surgical assessments to the 1-year post-surgery follow-up indicate a 'moderate' improvement in sound quality, this trend was consistent across most patients. However, variability in individual

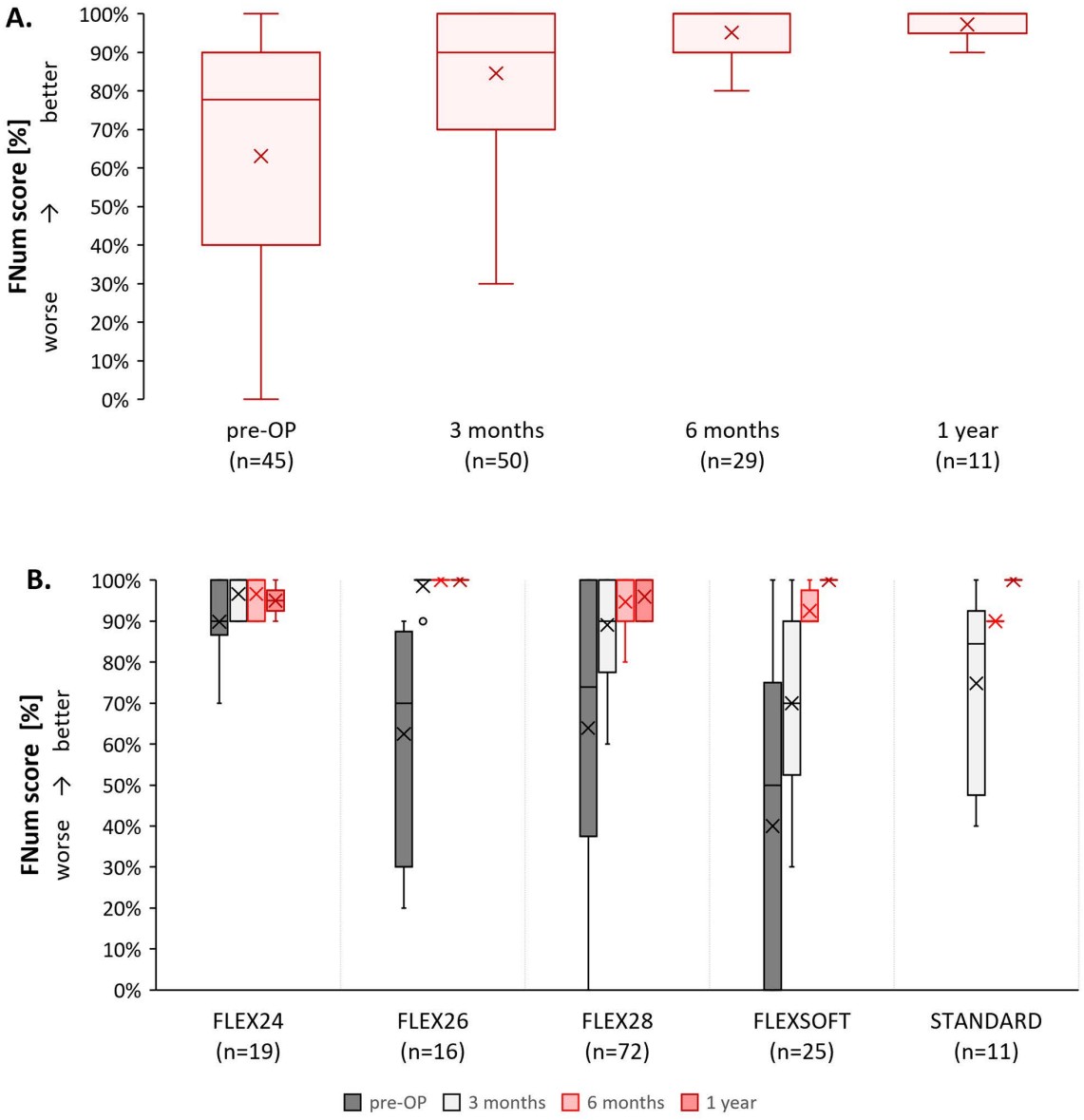

**Fig 4. Results on the Freiburg number test (FNum) [%] stratified by A) interval and B) electrode array and interval.** Note: if only an X is visible, only one measurement became available for the specified electrode array at the specific period. Small circles indicate outliers.

responses suggests that certain subgroups may experience differing levels of benefit, warranting further investigation.

## Safety

During the data collection period, 17 minor clinical events were reported in subjects using the SONNET 2 (EAS) audio processor. These events included headaches, migraines, mild pain around the implant site, and itchiness, all of which were reported to clinicians and documented in the data network. Additionally, 18 device deficiencies were recorded by participants using the SONNET 2 (EAS) audio processor.

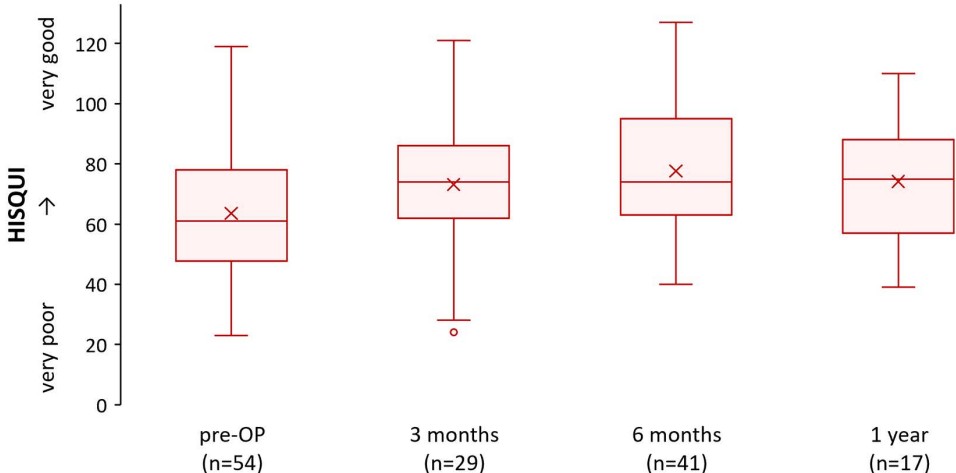

**Fig 5. Changes from pre-surgery to 1-year post-surgery on the HISQUI19.** The scores are shown in ascending order from very poor to very good. It should be noted that it is not possible to determine from the registry how many respondents already had a (contralateral) cochlear implant at the pre-surgical (pre-OP) assessment.

## Discussion

Five clinics (Munich, Oldenburg, Frankfurt, Heidelberg, and Bochum) participated in this analysis. The primary outcome of this investigation was to present the MEHS registry and demonstrate its functionality by extracting a pre-defined subsample of SONNET 2 or SONNET 2 EAS users. As such, this investigation was success: although not yet optimized, the MEHS registry facilitated a) consistent and reliable cross-cultural data collection across all age groups via an electronic platform and b) gathering extensive longitudinal datasets to provide evidence of device effectiveness, here the here the SONNET 2 (EAS).

The MEHS registry was created as a non-interventional systematic collection of routine clinical data from children and adults to help guide clinical practice. Clinical patient registries have been commonly used to collect uniform data on patients with specific diseases or conditions to guide clinical treatments and policies [25–29]. Electronic registries been previously created to collect mass information on routine clinical care that allows clinicians to improve standard care for CI users [12,18,19,30–37]. Additionally, several countries have already implemented specific CI registries [14], including Switzerland [38], France [33], Sweden, the Netherlands [14], and Germany [30]. A Germany-specific registry also allows for the capture of granular, high-quality, longitudinal patient-level data that is often underrepresented in pan-EU analyses. This complements other European registries by providing country-specific insights into outcomes in diverse care settings. Further, German regulatory and HTA (Health Technology Assessment) bodies often require localized evidence for reimbursement and policy decisions, further emphasizing the necessity of dedicated national data.

Other CI manufacturers have also recently published their experiences on the design, implementation, and management of their own international CI registry [31]. The publications generated from this registry, while important, are not comparable to the present study because they focused on specific patient demographics (i.e., children) and/or the effect of CI use on patient-reported outcomes measures; they did not assess speech understanding with specific arrays [e.g., 39–43].

The large amount of data collected within these electronic registries have allowed clinicians to make guided decisions on post-care after implantation. As previously mentioned, these electronic registries can provide data demonstrating overall improvements in speech perception and quality of life scores over time [18,31]. This, however, is true of CI registries in general. What distinguishes the MEHS from other CI registries is its emphasis on a more user-centered approach

to post-market surveillance and real-world data collection. Although manufacturer- and country-specific, it is designed to capture a broader ecosystem of hearing outcomes, surgical techniques, and device usage patterns, potentially including hybrid or bimodal users, where applicable. It collects a wide range of real-world data including patient-reported outcomes, audiological measures, and usage patterns over extended follow-up periods. The general phrasing of the MEHS registry`s objective allows for collection of clinically relevant data from current clinical practice on individuals affected by hearing loss, and for hypotheses generation. Thus far, 1682 participants have been enrolled in the MEHS registry over six years. In size, this is comparable to a previous CI registry, developed by Cochlear, which enrolled 1,500 participants over 10 years across 77 clinics [31]. The large number of participants, across multiple clinics, allows for extensive data availability on the demographics and condition characteristics of a much larger patient population. Registries, especially the current one, provide routine clinical data whereby hypothesis-driven research investigation could be created on individual or multiple variables, such as auditory performance, quality of life, and health-related utilities across different types of implantable and non-implantable hearing devices from MED-EL. The use of ENTstatistics within the MEHS registry offers a notable advantage by eliminating the necessity for duplicated data entry by clinicians, thereby reducing administrative burden. The theoretical premise involves the direct extraction of data from audiometers and the hospital's patient information system. Lastly, the MEHS is designed to align with EU data standards and GDPR compliance, allowing for harmonization with national health databases and the support of health economic analyses with Germany.

However, the use of the MEHS registry was not without its limitations, despite these limitations being overtly few. Firstly, neither the speech nor the QoL tests are obligatory for CI users, and this could result in a bias during the analysis of the data. It must be understandable that there is always a risk of participant selection bias in these types of registries, where some measurements and inclusion are not obligatory [44]. Despite the possibility of having selection bias, when conducting registry investigations, the collected data serve to guide clinical practice and are taking a more prominent role in overall clinical research. Second, CI users who have more complex and time-consuming conditions are less likely to be recruited to the registry, which has been demonstrated in a previous investigation using humans suffering from acute coronary syndrome [25]. Irregular inclusion bias could be present within the MEHS registry; however, this investigation focuses on best clinical practices regarding the use of the SONNET 2 (EAS). Thus, despite any bias that could be present, it is apparent that the SONNET 2 (EAS) improves hearing performance in adults suffering from severe to profound hearing loss. So, while 369 SONNET 2 (EAS) users could be included, results on speech tests, daily wearing time, etc., were available for a quarter or less of these users. While this still generated a relatively high n for a CI study, it highlights the continued difficulty in pooling data, even within a single country. A guideline ("Weißbuch") for CIs, which lists recommendations for structure, organization, equipment, qualifications, and quality assurance in the care of patients with a CI in Germany, is available to clinicians and may help standardize treatment [45]. All clinics perform CI aftercare following this guideline. Thus, while homogeneity is increasing, because clinics differ in their clinical practice, data derived from it will always be prone to higher rates of heterogeneity than those derived from planned studies. One potential remedy for this is the adherence to the CI-ICF assessment protocol recently developed by Andres et al. [46], which proposes a loose testing scheme. The authors are discussing further ways to increase homogeneity. Fourth, concerning HISQUI19 scores, it was not possible to determine from the registry the number of respondents that already had a (contralateral) CI or other hearing device at the pre-surgical assessment. This may have contributed to the relatively high variability in results. Lastly, despite the introduction of the ENTstatistics platform, which was devised to streamline the collection of anonymous routine clinical data, the execution of the registry encountered several obstacles. The initial integration of ENTstatistics into the hospital's IT system posed significant challenges. Additionally, the automated transmission of data for speech tests from certain types of audiometers was absent, mandating manual intervention by clinicians. Given the temporal limitations faced by clinicians in a clinical setting, this manual data entry procedure was not consistently carried out. This is reflected in the smaller volume of post-operative speech test data available compared to the initial dataset. Furthermore, discrepancies emerged due to variations in the methodologies employed by different clinics for entering routine clinical data,

resulting in heterogeneous data input. An additional complication arose from instances where entries were erroneously attributed; for instance, measurements performed with CIs were occasionally mislabeled as being executed with hearing aids. These encountered obstacles presented considerable challenges, which, over the years, have been progressively mitigated. The obstacles that arose from this registry confirmed that cochlear implantation and its aftercare are a diverse and heterogeneous landscape. CI treatment requires a complex and interdisciplinary medical treatment. Our example of a clinical patient registry shows that the success of registries heavily depends on the consistent participation and compliance of various clinical centers. Variability in the level of engagement and adherence to standardized protocols across centers can impact the quality and completeness of the data. Differences in clinical practices, surgical techniques, patient demographics, and reporting standards across centers could result in variability that may impact the homogeneity of the results. Despite aiming for a comprehensive data collection from multiple centers, this registry shows the challenges related to the homogeneity of the data and ultimately to the usability of the collected data.

The secondary objective of this manuscript was to demonstrate the utility of the MEHS registry in collecting routine clinical data on the performance, effectiveness, and safety of the SONNET 2 (EAS) audio processor. In assessing the processor's performance, our analysis revealed that the registry data for both the FMS and FNum tests were consistent with findings from previous studies [20,23,47–52]. For instance, a study by Kurz and colleagues [20] similarly reported no significant difference in FMS test results with the SONNET 2 (EAS) compared to the older SONNET (EAS) model. Furthermore, aligning with our findings, other studies have demonstrated improvements in the FMS scores from pre- to post-surgery [53–56]. Additionally, and comparable to the FMS results, a study by Zeh and Baumann [57] showed improvements in FNum scores between pre- and post-surgery following 3–5 weeks of inpatient hearing therapy [57]. The comparable results are clear examples that the data extracted from this registry were accurate. One conclusion that may be drawn from the results is that the FNum test is probably too easy for CI users, and so a more difficult test may yield more helpful results. Lastly, the compiled results within the registry demonstrate that the SONNET 2 (EAS) audio processors are effective and safe to use. The patients wore the device throughout the day, indicating that the devices worked effectively and were comfortable to wear [52].

In conclusion, the MEHS registry has shown, based on the data compiled, that it is an effective method of storing data from a large number of clinics and that it is functional for extracting pre-defined subsamples (e.g., specific devices). This data can be used to generate hypotheses and provide results that may be used to assist with patient rehabilitation treatment. Unlike the strictly defined cohorts with specific inclusion and exclusion criteria in clinical studies, registries provide a more unbiased and realistic picture of real-world data. Moreover, the registry showed that the SONNET 2 (EAS) audio processors perform well, are effective, and are safe to use.

## Supporting information

**S1-3 Tables. Raw data used in the manuscript.**
(DOCX)

## Acknowledgments

The authors would like to thank all the CI users who graciously gave their time volunteering to be a part of this registry. The authors would like to thank Alexander Hansen and Michael Todd (both MED-EL) for writing and editing a draft of this manuscript. AI tools were not used in the creation of this manuscript.

## Author contributions

**Conceptualization:** Uwe Baumann, Tobias Weissgerber, Andreas Radeloff, Vera Lohnherr, Stefan Dazert, Christiane Völter, Ilona Anderson, Joachim Müller.

**Data curation:** Karin A. Koinig, Magdalena Breu.

**Formal analysis:** Karin A. Koinig, Magdalena Breu.

**Investigation:** Uwe Baumann, Tobias Weissgerber, Andreas Radeloff, Jasmine Rinnofner, Vera Lohnherr, Stefan Dazert, Christiane Völter, Ilona Anderson, Joachim Müller, Daniel Polterauer.

**Writing – review & editing:** Uwe Baumann, Tobias Weissgerber, Andreas Radeloff, Karin A. Koinig, Magdalena Breu, Jasmine Rinnofner, Vera Lohnherr, Stefan Dazert, Christiane Völter, Ilona Anderson, Joachim Müller, Daniel Polterauer.

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
