## [Decision Letter · Decision Letter 0]

20 May 2025

Dear Dr. Baumann,

Thank you for submitting your manuscript to PLOS ONE. After careful consideration, we feel that it has merit but does not fully meet PLOS ONE’s publication criteria as it currently stands. Therefore, we invite you to submit a revised version of the manuscript that addresses the points raised during the review process.

We look forward to receiving your revised manuscript.

Kind regards,

Renato S. Melo, PhD

Academic Editor

PLOS ONE

Journal Requirements:

This study was funded by MED-EL Elektromedizinische Geräte G.m.b.H.

5. Please remove all personal information, ensure that the data shared are in accordance with participant consent, and re-upload a fully anonymized data set.

6. We note you have included a table to which you do not refer in the text of your manuscript. Please ensure that you refer to Table 3 in your text; if accepted, production will need this reference to link the reader to the Table.

Reviewers' comments:

Reviewer's Responses to Questions

**Comments to the Author**

1. Is the manuscript technically sound, and do the data support the conclusions?

Reviewer #1: Yes

Reviewer #2: Yes

2. Has the statistical analysis been performed appropriately and rigorously?

Reviewer #1: Yes

Reviewer #2: Yes

3. Have the authors made all data underlying the findings in their manuscript fully available?

Reviewer #1: Yes

Reviewer #2: Yes

4. Is the manuscript presented in an intelligible fashion and written in standard English?

Reviewer #1: Yes

Reviewer #2: Yes

Reviewer #1: This is a well-executed study with strong real-world applicability. Minor refinements, such as enhancing clarity, statistical nuance, and comparative context, could further elevate its impact for clinicians, policymakers, and CI manufacturers. Below is a detailed review with strengths, areas for improvement, and suggestions for refinement.

1. Abstract & Plain Language Summary

•Abstract:

oConsider adding effect sizes (e.g., mean improvement in speech scores) for greater impact.

oClarify if bilateral vs. unilateral users showed differential outcomes.

•Plain Language Summary:

oSimplify technical terms (e.g., "Freiburg tests" → "speech understanding tests").

o Emphasize why registries matter (e.g., "larger datasets lead to better treatment guidelines").

2. Introduction

•Contextualize the SONNET 2 (EAS):

o Briefly explain how it differs from previous models (e.g., SONNET 1) to justify its evaluation.

•Registry Rationale:

o Strengthen the argument for Germany-specific data (e.g., how this complements other EU registries).

3. Methods

•Data Collection:

o Clarify how missing data (e.g., post-op tests) was handled statistically.

o Justify Mann-Whitney U tests for non-normal distributions (add Shapiro-Wilk test results if available).

•Subgroup Analysis:

oElectrode arrays (FLEX28, FLEXSOFT, etc.) had small samples—note if results are exploratory.

4. Results

•Speech Tests:

o Ceiling Effect: The FNum test’s simplicity (high post-op scores) may limit sensitivity—discuss implications.

o Electrode Variability: FLEXSOFT users had lower FMS scores (43.3%) vs. FLEX28 (62.1%). Is this clinically significant?

•HISQUI19:

o Highlight variability in self-reported benefit (Fig. 5). Could pre-existing contralateral CIs explain this?

5. Discussion

•Registry Challenges:

o Expand on solutions for data homogeneity (e.g., centralized training for clinics).

o Compare MEHS vs. other CI registries (e.g., Cochlear’s registry) to highlight best practices.

•SONNET 2 (EAS) Performance:

o Contrast with competitor devices (e.g., Cochlear’s Nucleus, AB’s Marvel) if literature permits.

6. Language & Flow

•Avoid Redundancy:

oThe Plain Language Summary and Introduction overlap—consider merging or trimming.

•Active Voice:

o E.g., "We observed" → "Our analysis showed" for conciseness.

Reviewer #2: The study is a prospective, multicenter registry of patient outcomes for CI users of all ages has been developed. This study highlights the use of the MED-EL Hearing Solutions (MEHS) registry, with the primary aims of; facilitating consistent and reliable cross-cultural data collection across all age groups via an electronic platform, and gathering extensive longitudinal datasets to provide evidence of device effectiveness. The secondary aim of this investigation was to collect routine clinical data on the performance, effectiveness, and safety of the SONNET 2 and SONNET 2 EAS audio processors. Five clinics in Germany (Munich, Oldenburg, Frankfurt, Heidelberg, and Bochum) until 31st May 2023, participated in this analysis.

Overall, it is a well-thought out and executed research manuscript capable of contributing to the body of knowledge especially due to its ample sample size (426 ears, 57 bilateral users). However, I have the following observations,

1. Introduction: Some of the refences appear to have some typo. Looking at Pg 4, Line 105-107 for example, Olusanva & Newton, 2007 appears wrongly spelt when compared with the correct reference in literature.

2. Methods: The statement made in the methodology section Pg 9, Line 105-107, ‘Device deficiencies were classified based on the severity of the issues to the device and its potential impact on patients’ safety’ appears difficult to validate especially in patient’s not currently on admission. Hence relying on patient- reported results to validate clinical safety status might not be accurate.

The reported findings of ‘17 minor clinical events were reported in subjects using the SONNET 2 (EAS) audio processor’ being self-reported, may not be correctly termed as being, no new/unresolved safety concerns reported.

3. The phrasing of the language used in the entire document could benefit from a more professional grammatical input.

4. I found the title, study design, referencing, statistics, figures and tables quite satisfactory.

**Do you want your identity to be public for this peer review?** For information about this choice, including consent withdrawal, please see our Privacy Policy

Reviewer #1: No

Reviewer #2: **Yes: ** Stephen O Adebola

---

## [Author Response · Author response to Decision Letter 1]

7 Jul 2025

Editor

1. Please ensure that your manuscript meets PLOS ONE's style requirements, including those for file naming. The PLOS ONE style templates can be found at https://journals.plos.org/plosone/s/file?id=wjVg/PLOSOne_formatting_sample_main_body.pdf

& https://journals.plos.org/plosone/s/file?id=ba62/PLOSOne_formatting_sample_title_authors_affiliations.pdf

We have reformatted the manuscript to conform with PLOS ONE’s style requirements. Please note, we did not use track changes for this (because the changes would be overwhelming & do not affect intellectual content).

There are no grant numbers.

This study was funded by MED-EL Elektromedizinische Geräte G.m.b.H.

We have added

“The funders assisted in study design, data collection and analysis, and preparation of the manuscript. Decision to publish was a joint decision between all authors.”

to the cover letter & to the Funding Sources section at the end of the manuscript.

We have included raw data for 1) the values behind the means, standard deviations and other measures reported & 2) the values used to build graphs.

“The points extracted from images for analysis.” is not relevant in this manuscript.

5. Please remove all personal information, ensure that the data shared are in accordance with participant consent, and re-upload a fully anonymized data set.

Data are anonymized at the source.

6. We note you have included a table to which you do not refer in the text of your manuscript. Please ensure that you refer to Table 3 in your text; if accepted, production will need this reference to link the reader to the Table.

Table 3 is referred to twice in the section “Freiburger monosyllabic speech test” & in the section “Freiburger number test”

Additionally, please note that we have corrected the descriptive results to median and IQR. When the n was 3 or lower, maximum and minimum scores are given. Please see lines appx 182-186.

NEXT PAGE – REVIEWER 1

REVIEWER 1

Reviewer #1: This is a well-executed study with strong real-world applicability. Minor refinements, such as enhancing clarity, statistical nuance, and comparative context, could further elevate its impact for clinicians, policymakers, and CI manufacturers. Below is a detailed review with strengths, areas for improvement, and suggestions for refinement.

Abstract & Plain Language Summary

1. Abstract

a. Consider adding effect sizes (e.g., mean improvement in speech scores) for greater impact.

The median scores at pre-op and 1 yr post-op have been added. We would prefer to not add the interquartile scores but we think it would be more information than necessary for an abstract and would be the text less readable.

b. Clarify if bilateral vs. unilateral users showed differential outcomes.

This was out of the scope of the study and was therefore not assessed, although it could certainly be a topic for a future study.

In addition, we excluded binaural tests to focus on the implant (binaural would also include bimodal tests).

2. Plain Language Summary

a. Simplify technical terms (e.g., "Freiburg tests" → "speech understanding tests").

Could you please give an example? Freiburg tests are not mentioned in the PLS. We could not find other technical terms that would likely be not understood by ENTs.

b. Emphasize why registries matter (e.g., "larger datasets lead to better treatment guidelines").

A sentence to this end has been added.

3. Introduction

a. Contextualize the SONNET 2 (EAS):

i. Briefly explain how it differs from previous models (e.g., SONNET 1) to justify its evaluation.

We added a new section in the Methods (lines 76-83) to briefly describe how the SONNET 2 differs from previous models. We put it in the Methods because we would like the Introduction to focus on registries and their potential.

b. Registry Rationale:

i. Strengthen the argument for Germany-specific data (e.g., how this complements other EU registries)

We added this in the Discussion, where we mention registries in other countries (appx lines 319-324). The reason MED-EL established the MEHS registry is that it does not have access to the German DCIR registry, as per their policy.

4. Methods

a. Data Collection:

i. Clarify how missing data (e.g., post-op tests) was handled statistically.

ii. Justify Mann-Whitney U tests for non-normal distributions (add Shapiro-Wilk test results if available).

The above 2 comments have been addressed in the “Statistical analysis” section (lines appx 181-185).

b. Subgroup Analysis:

i. Electrode arrays (FLEX28, FLEXSOFT, etc.) had small samples—note if results are exploratory.

This is addressed in the Results “Freiburg monosyllabic speech test” (line appx 230) where we state that inferential statistics could only be applied for the FLEX28. Results for the other arrays are presented descriptively. We added “inferential” to make this clearer. That FLEX28 had a higher n than other arrays no doubt reflects real life usage in Germany. In countries, one might expect a higher use shorter arrays due to more cases of cochlear malformation.

5. Results

a. Speech Tests:

i. Ceiling Effect: The FNum test’s simplicity (high post-op scores) may limit sensitivity—discuss implications.

We now state in the Discussion section (appx. Iines 430-2) that the FNum is probably too easy for CI users and so using a more difficult test would be helpful. The FNum test is nonetheless helpful because the standard OLSA test cannot usually be used at preOP.

ii. Electrode Variability: FLEXSOFT users had lower FMS scores (43.3%) vs. FLEX28 (62.1%). Is this clinically significant?

We would need to have a larger sample size for FLEXSOFT to compare these two electrode arrays. Sample size was also the reason why only FLEX28 was tested with inferential statistics.

b. HISQUI19:

i. Highlight variability in self-reported benefit (Fig. 5). Could pre-existing contralateral CIs explain this?

We address this in lines appx. 391-93.

6. Discussion

a. Registry Challenges

i. Expand on solutions for data homogeneity (e.g., centralized training for clinics)

We expanded on this in lines appx 385-90. The introduction of the Weißbuch should help with homogeneity. Additionally, using the CI-ICF (Andries et al., in the references) would help but as the data are derived from each clinic’s routine, improving homogeneity is certainly a challenge. The authors will be discussing this together in the near future.

ii. Compare MEHS vs. other CI registries (e.g., Cochlear’s registry) to highlight best practices.

We have added a paragraph to address this, please see lines appx 338-366.

b. SONNET 2 (EAS) Performance:

i. Contrast with competitor devices (e.g., Cochlear’s Nucleus, AB’s Marvel) if literature permits.

Direct comparisons between cochlear implant systems from different manufacturers—such as MED-EL’s SONNET 2, Cochlear’s Nucleus series, or Advanced Bionics’ Marvel platform—are limited in the literature due to variability in clinical protocols, patient populations, and outcome measures. Most published studies avoid head-to-head comparisons, recognizing the challenges of establishing equivalency across diverse technological ecosystems and proprietary algorithms. Furthermore, for an equal comparison, we would probably need to compare two sets of real-world evidence.

As for registries, the papers we can find from Cochlear’s IROS (and P-IROS) registry assess specific patient groups (i.e., children in the P-IROS) or specifically focus on patient-reported outcomes measures, so aren’t comparable to the present study.

We’ve added information to this effect to the Discussion. See lines appx 333-336.

c. Language & Flow

i. Avoid Redundancy:

1. The Plain Language Summary and Introduction overlap—consider merging or trimming.

This is intentional because the Plain Language Summary and the Introduction are for different audiences so are independent of each other.

ii. Active Voice:

1. E.g., "We observed" → "Our analysis showed" for conciseness.

Each instance of “we observed” has been edited out.

NEXT PAGE REVIEWER 2

REVIEWER 2

Overall, it is a well-thought out and executed research manuscript capable of contributing to the body of knowledge especially due to its ample sample size (426 ears, 57 bilateral users).

However, I have the following observations,

1. Introduction: Some of the refences appear to have some typo. Looking at Pg 4, Line 105-107 for example, Olusanva & Newton, 2007 appears wrongly spelt when compared with the correct reference in literature.

This has been corrected in the Introduction (correct spelling Olusanya). We could not find other references that have typos. All have been reformatted as per PLOSOne requirements.

2. Methods: The statement made in the methodology section Pg 9, Line 105-107, ‘Device deficiencies were classified based on the severity of the issues to the device and its potential impact on patients’ safety’ appears difficult to validate especially in patient’s not currently on admission. Hence relying on patient- reported results to validate clinical safety status might not be accurate.

We have added a sentence to the end of the “Safety” section to address this, see lines appx 113-5.

The reported findings of ‘17 minor clinical events were reported in subjects using the SONNET 2 (EAS) audio processor’ being self-reported, may not be correctly termed as being, no new/unresolved safety concerns reported.

This sentence in the Abstract has been changed to “Safety was assessed as per participant self-report: 17 minor clinical events were reported, none were new or unresolved”

3. The phrasing of the language used in the entire document could benefit from a more professional grammatical input.

All content in the document re-edited and reviewed by a native English speaker and professional medical writer.

---

## [Decision Letter · Decision Letter 1]

9 Oct 2025

MED-EL Hearing Solution Registry: An examination of the strengths and limitations of a cochlear implant registry

PONE-D-25-18559R1

Dear Dr. Baumann,

We’re pleased to inform you that your manuscript has been judged scientifically suitable for publication and will be formally accepted for publication once it meets all outstanding technical requirements.

Kind regards,

Andreas Buechner

Academic Editor

PLOS ONE

Additional Editor Comments (optional):

Reviewers' comments:

Reviewer's Responses to Questions

**Comments to the Author**

Reviewer #1: All comments have been addressed

Reviewer #2: All comments have been addressed

2. Is the manuscript technically sound, and do the data support the conclusions?

Reviewer #1: Yes

Reviewer #2: Yes

3. Has the statistical analysis been performed appropriately and rigorously?

Reviewer #1: Yes

Reviewer #2: Yes

4. Have the authors made all data underlying the findings in their manuscript fully available?

Reviewer #1: Yes

Reviewer #2: Yes

5. Is the manuscript presented in an intelligible fashion and written in standard English?

Reviewer #1: Yes

Reviewer #2: Yes

Reviewer #1: Your manuscript is well-structured, Well-defined and appropriate statistical controls. It worse publications

Reviewer #2: Found the responses to the observations made to be acceptable and would recommend the manuscript be accepted in current format

**Do you want your identity to be public for this peer review?** For information about this choice, including consent withdrawal, please see our Privacy Policy

Reviewer #1: **Yes: ** Mohammed Elrabie Ahmed

Reviewer #2: **Yes: ** Adebola, Stephen Oluwatosin

---

## [Editor Report · Acceptance letter]

PONE-D-25-18559R1

PLOS ONE

Dear Dr. Baumann,

I'm pleased to inform you that your manuscript has been deemed suitable for publication in PLOS ONE. Congratulations! Your manuscript is now being handed over to our production team.

Kind regards,

on behalf of

Andreas Buechner

Academic Editor

PLOS ONE